# Security and Safety Concerns in Air Taxis: A Systematic Literature Review

**DOI:** 10.3390/s22186875

**Published:** 2022-09-12

**Authors:** Isadora Garcia Ferrão, David Espes, Catherine Dezan, Kalinka Regina Lucas Jaquie Castelo Branco

**Affiliations:** 1Institute of Mathematics and Computer Sciences, University of São Paulo, Ave. Trabalhador São-Carlense, 400, São Carlos 13564-002, SP, Brazil; 2Lab-STICC, Université de Bretagne Occidentale (UBO), 29200 Brest, France

**Keywords:** air taxi safety, air taxi security, air taxi architecture

## Abstract

Different from traditional transport systems, such as cars or trains, which are limited by land transit space, flying cars (such as UAS, drones, and air taxis) do not occupy space with traffic. They have a degree of freedom in space and time, smaller displacement, and consequently, less stress for their users. Large companies and researchers around the world are working with different architectures, algorithms, and techniques to test air taxi transport to serve a significant proportion of people safely and autonomously. One of the main issues surrounding the diffusion of air taxis is safety and security, since a simple failure can lead to the loss of high-value assets, loss of the vehicle, and/or injuries to human lives, including fatalities. In this sense, despite significant efforts, the literature is still specific and limited regarding air taxi safety and security. Therefore, this study aimed to carry out an extensive systematic literature review of the main modern advances in techniques, architectures, and research carried out around the world focused on these types of vehicles. More than 210 articles from between 2015 and January 2022 were individually reviewed. In addition, this study also presents gaps that could serve as a direction for future research. As far as the authors are aware, no other study performs this type of review focused on air taxi safety.

## 1. Introduction

Urban planning and mobility needs to fit the increase of the population in big centers around the world to ensure quality of life and to address sustainable economic development, safe transportation infrastructure, and current long-term demands for future generations. It is estimated that in 2030, the population will reach 8.5 billion [1]. Current urban mobility public policies, especially in bigger centers, are facing obstacles to accommodating larger populations. Moreover, one of the main symptoms of poor urban mobility conditions is congestion. According to [2], congestion causes anxiety, insecurity, and impotence, generating chronic stress due to repetitive situations. In addition to congestion, there is population swelling and the expansion of cities, with increasing peripheral areas, which the government neglects in relation to mobility policies. Urban mobility problems also affect the economy due to lack of mobility and loss of productivity [3,4].

As a way to solve the problems related to urban mobility, urban air mobility (UAM) is an emerging form of air transport proposed by NASA and Uber [5,6] within the last five years. UAM envisions a safe, sustainable, and affordable air transport system for passenger mobility, goods delivery, and emergency services within or across metropolitan areas. Unlike traditional transport systems such as cars or rail trains, which are limited by land transit space, flying cars (such as UAS, drones, and air taxis) do not occupy spaces with traffic. They have a degree of freedom in space and time, less displacement, and consequently, create less stress for their users.

One of the main bets of companies and research centers for UAM are air taxis, also known as flying cars or electric vertical take-off and landing aircraft (eVTOLs) that will transport loads and people in urban airspace efficiently and with high speed [7,8,9]. Air taxis will be responsible for meeting on-demand transport needs, connect important transport nodes (for example, the airport with the city center), and provide fast travel between train stations or across rivers and lakes, operating 24 h a day [10]. But all this will only be possible thanks to drone ports (also called vertiports or skyports) that will be arranged in various cities, allowing easy access, transfer to other modes of transport, and technical support such as battery charging and maintenance for air taxis.

Studies indicate that cities will have sophisticated urban air mobility in the near future, with thousands of air taxi flights around the world [11,12,13,14,15,16,17]. However, air taxis have been the subject of studies by companies and centers worldwide. This is because the ability to manage many of these aircraft in congested urban areas presents several unexpected challenges during travel. These studies range from the social acceptance of these aerial vehicles, noise problems, types of architecture, and safety and autonomy issues.

One of the main concerns surrounding the diffusion of air taxis is the safety and security aspects, since a simple failure can lead to the loss of high-value assets, loss of vehicles, and/or injury to human lives. Despite significant efforts, the literature on air taxi safety and security is still specific and limited. Therefore, this study aimed to carry out an extensive systematic literature review (SLR) to present the main modern advances with respect to the techniques, architectures, and research carried out worldwide focused on these types of vehicles. This study also has gaps that can serve as a guide for future research.

### Related Studies

It is possible to find literature review works on some air taxi segments, as can be seen in Table 1, such as for urban air mobility [18], review of services [19], demand for services [20], and others. However, despite the great dangers of a simple safety or security failure, as far as we are aware, there is no other work reviewing air vehicle safety and security.

## 2. Systematic Literature Review Method

This study aimed to carry out an extensive systematic literature review (SLR) seeking to present the main modern advances in the techniques, architectures, and research carried out around the world focused on these types of vehicles. In addition, this study also presents gaps that could serve as a direction for future research.

An SLR aims to identify, analyze, and interpret all available evidence regarding a particular research question in an unbiased and repeatable manner. Through this technique, it is possible to summarize possible proof of a technology, identify gaps to suggest areas that should be investigated, clarify the state-of-the-art for the positioning of new research, and examine empirical results that prove or refute a given theoretical hypothesis or that assist in the creation of new hypotheses [22]. As mentioned in the previous paragraph, this review focused on evidence from the literature for air taxis, mainly on the architectures, algorithms, and techniques that deal with safety and security.

The systematic review process presented in this section was carried out with the collaboration of the online tool Parsifal. Parsifal provides all necessary support to execute the systematic review protocol in a formalized, stepped way, including review objectives, research questions, keywords, synonyms, inclusion and exclusion criteria, data extraction, and others [22,23].

This review was divided into four stages. During Step 1, the plan consisted of the definition of objectives, search sources, criteria for inclusion or exclusion of studies, the language of the studies, search key, etc. In Step 2, the execution was carried out, where the studies were taken from the predefined sources during Step 1 and placed in Parsifal. Still, in this same stage, reading of the titles and abstracts were also carried out, applying the inclusion and exclusion criteria. However, in this step, they are read entirely and not just the title and abstract. Finally, in Step 2, the results were synthesized.

### 2.1. Step 1: Planning

The main objective of this review was to identify techniques, architectures, and algorithms used to diagnose and mitigate attacks on air taxis. Research questions, search sources, search strings, and inclusion and exclusion criteria were defined during the planning phase. Among the research questions, the following were defined:What are the main techniques that have been investigated to provide security and safety support for air taxis?What types of tests (e.g., simulation, databases, etc.) are used to test security and safety techniques in air taxis?What is missing for air taxis to be implemented in everyday life?What gaps exist in safety and computer security in air taxis?

### 2.2. Search Strategy

The language determined for the review of the works was English, the language adopted worldwide for scientific research. The following criteria were established to select the sources used in the review: coverage, availability, versatility to export the results, and usability. Only studies with full-text access were reviewed for availability. For versatility, sources were determined that exported the articles automatically. Finally, accepted search engines were also considered using usability criteria that are easy to understand and operate. Based on the above criteria, the search sources were:IEEE Xplore Digital Library [24];ACM Digital Lybrary [25];Scopus [26];Web of science [26].

For the systematic review, the following keywords and their synonyms were chosen:Security: safety and security;Vehicles: air taxi and eVTOL;Attacks: attacks and cyber attacks.

Further, at this stage, the inclusion and exclusion criteria were defined, seeking to guarantee the review’s credibility and to maintain focus on the selection. The inclusion criteria for the identified studies were:Studies that address aspects of safety and/or security in air taxis;Studies that use techniques and/or algorithms of safety or security in flying cars;Studies that do not address safety or security but address innovations for flying vehicles.

The defined exclusion criteria were:Studies that do not have a clear structure of objectives and results;Studies that have no relevance to the main theme;Studies that address flying vehicles but do not address safety or security aspects;Incomplete studies;Studies that are not in English language;Studies that are not available for download.

At the end of this phase, two experts evaluated the protocols defined during planning to verify the consistency of the definitions. At the end of the planning process and before starting the execution of the systematic review, it is necessary to validate the protocol with specialists, thus certifying quality during the remainder of the search. A software engineering specialist who focuses on systematic review and an autonomous vehicle specialist who focuses on vehicle safety were invited for the evaluation. After that, all suggestions were considered before starting the next step.

### 2.3. Step 2: Execution

The execution stage begins with the process of identifying the primary studies using the search strategies defined in the planning phase. For this, in the first step, it is necessary to create the search strings. The search strings were designed from the keywords defined earlier in the planning phase of the review protocol. The second step consists of preliminary article selection by reading the titles and abstracts and assessing whether the works fit the inclusion and exclusion criteria. In the third step, the final selection occurs through the complete reading of the selected articles and data extraction. Finally, in the fourth step, the data are synthesized and analyzed.

### 2.4. Search String

The search string was created respecting the specifications of each database, using “AND” and “OR” operators to group synonyms. The search strings created with the terms in English were:IEEE: (((“Full Text Only”:“Safety” OR “Full Text Only”:“Security”) OR (“Full Text Only”:“attacks” OR “Full Text Only”:“cyber attacks”)) AND (“Full Text Only”:“air taxi” OR “eVTOL”)).Web of science: TS = (((“Safety” OR “Security”) OR (“Attack” OR “cyber attacks”)) AND (“air taxi” OR “eVTOL”)).SCOPUS: TITLE-ABS-KEY(((“safety” OR “security”) OR (“attacks” OR “cyber attacks”)) AND (“air taxi” OR “eVTOL”)) AND PUBYEAR > 2015.ACM: [[Full Text: “safety”] OR [Full Text: “security”]] OR [[Full Text: “attacks”] OR [Full Text: “cyber attacks”]] AND [[Full Text: “air taxi”] OR [Full Text: “evtol”]].

### 2.5. Study Selection

After performing the search in repositories with the search strings, defining a period from January 2015 to January 2022, a total of 213 studies were identified. Their information was exported from the databases in Bibtex format and imported on the Parsifal platform. The database with the most significant number of articles related to the topic was IEEEXplore Digital Library, followed by the SCOPUS, Web of Science, and ACM Digital Library, as its possible to see in Figure 1. One hundred and fifteen articles were identified in IEEEXplore Digital Library (53.3%), seventy-two in SCOPUS (34.2%), twenty in Web of Science (9.3%), and only six in Scopus (2.8%).

The first selection was made by reading the titles and abstracts, identifying the inclusion and exclusion criteria. The articles accepted were those that met at least one of the inclusion criteria, and the rejected articles were those that met at least one exclusion criterion. In the end, 178 articles were accepted, and 35 were rejected. The distribution of the selected articles in each of the bases can be observed in the graph of Figure 2. The highest number of articles were found in the IEEExplorer Digital Library database, followed by SCOPUS.

After primary study selection, each accepted article was read in its totality and re-matched to an inclusion or exclusion criteria. During the full reading, summaries were generated as results. The number of articles accepted in the final selection stage can be seen in Figure 3. The highest number of articles found and consequently accepted were on IEEE, with a total of 41 studies selected, followed by SCOPUS, with 13 accepted articles, Web of Science, with 7, and only 2 from ACM.

Among the accepted articles, the criterion that received the highest level of acceptance was that of studies that use techniques and/or algorithms of safety or security in flying cars. Regarding the exclusion criteria, the most significant was the criterion of studies that have no relevance to the central theme. Most of them did not have characteristics relevant to the main focus of the research. In the last step of the systematic review, a discursive synthesis of results was produced to expose the results found and the research gaps. This synthesis is presented in Section 3.

## 3. Results and Discussion

Despite not being a new concept, studies that address air taxis are still scarce in the literature, and only 213 works in the last seven years were found in the worldwide literature reference bases. The categories most found in the literature and which will be discussed in the next sections are techniques for preventing and predicting accidents. Other works focus on the social acceptance of these types of vehicles through interviews with experts and the community in general. In addition, some proposals for safe architectures for air taxis are found in the literature and are discussed throughout the next sections.

### 3.1. Social Acceptance

Despite not being the focus of this review, the social acceptance of potential users, whether pilots or future users, is a recurring topic when reading studies [9,14,27,28,29,30,31,32]. According to [9], one of the existing research gaps in air taxis is in the technical, political, and legal fields concerning the illicit use of these vehicles beyond their programmed functions, since most of the technical efforts focus on physical integration functions.

Different studies have been found, from social acceptance of noise [27], climatic events [28], and interior design [29], to questions about sustainability and social and economic impacts [14].

An example of a study on social acceptance is presented by [27]. According to the researchers, the residents of future cities that will receive air taxis are seen as interested parties in urban air mobility, not only those users who will use the vehicle. This is because living under or close to an air taxi route will change the routine of these residents, for example, through the noise. In this study, the authors discuss essential aspects of the social acceptance of air mobility. In [28], the potential climatic challenges and public approval for operations in adverse conditions are presented. A general population survey of five cities assessed a seasonal and diurnal climatological analysis of adverse conditions using historical observations at operational altitudes and public perceptions of climate-related social barriers. Respondents were asked about their views on flying in a small aircraft in a variety of adverse weather conditions. The study shows the insecurity of the population in the face of adverse conditions.

Despite the wide variety of studies on social acceptance, the studies focus on theoretical interviews. There is still a lack of studies relating social acceptance with practical implementation, whether in software or hardware, through the recommendations of future users. In other words, practical tests or simulations using accepted concepts or proposals suggested by the interviewee’s theories still lack practical implementation through suggestions from future passengers.

### 3.2. Accident Prevention

As expected, accident prevention is the main reason that guides the works found in the literature. As shown in Table 2, the reasons for air taxi accidents can vary from human errors, equipment failures, lack of data, weather factors, and attacks, among others.

According to [34], most air vehicle accidents involve weather events or pilot errors. These data coincide with the results found during this review, since climate and human problems are part of the most recurrent categories found with theoretical and practical solutions. Further, according to [34], aviation accident and incident data and reports maintained by various agencies indicate that approximately 80% of accidents can be attributed to pilot error. The relationship between human errors and accidents emphasizes the importance of air taxis being composed of autonomous systems, since these vehicles will be exposed to adverse, random events and need to continually adapt regardless of the problems, aerial obstacles, or failures encountered during the route. However, for these vehicles to act autonomously, a series of analyses are required to avoid operational risks, such as physical security, cyber risks, traffic management, airports, collisions, etc.

Autonomy is linked to the entire vehicle process, not only at system levels but also in identifying failures in parts. For example, due to fatigue, wear, corrosion, or maintenance, problems can go unnoticed to the human eye but are fatal in the accident case. In [40,41], the authors present a technique named NCI, responsible for determining the priority of nodes in the network through an index, thus guaranteeing the quality of safety for the vehicle equipment. In their current state, the nodes receive a manual score determined by humans. Despite efforts with the NCI, the authors themselves mention the need for a mathematical formula that can automatically assign the data.

Two other important points highlighted in the literature are the climate and the absence of data. According to [27], aircraft operators expressed, after a series of interviews, major concerns about the information from stations not being timely enough (some data only available hourly) and far from their mission locations (typically ASOS are deployed only at major airports). According to the authors, the primary need for operators is the availability of detailed information about the wind during the critical phases of take-off and landing, unexpected wind gusts in urban canyons, and high-resolution spatial and temporal information about the location of thunderstorms, hail, and rain events to help make decisions about route selection, detour planning, and shelter for passengers and vehicles. Another fact mentioned in the literature is the absence of systems that can identify the weather, such as winds, on sunny days. In other words, it is not enough to provide general data, but complete access to real-time climate data is required. Meeting all current climate needs will require engineers and scientists to not only implement observational systems, but also databases that gather meteorological data from aircraft sensors to initialize numerical prediction models.

All studies found in the literature that relate to accidents due to attacks are focused on GPS, for example, GPS spoofing and jamming [35,40,42,46,48,49]. One of the reasons that may justify this is the fact that GPS is the most recurrent target of attacks on civilian air vehicles [40], since these vehicles do not have encryption, unlike military vehicles. These attacks, when successful, can lead to partial or total loss of the aerial vehicle. Another essential point to be highlighted is that only three works are concerned with safety and security together [40,46,50]. One of the concerns in the design and development of air vehicles is to ensure safety requirements. However, as these vehicles communicate with external entities, architectures designed to provide safety requirements may have security gaps, and security requirements may have safety gaps. Therefore, it is necessary to investigate and address safety and security in building architectures.

In relation to all the flaws presented in Table 2, some works in the literature present efforts to overcome safety or security problems through techniques and equipment to avoid or minimize these accidents [40,48,49,51,52,53,54,55,56,57], as can be seen in Table 3.

Some of the equipment presented for air taxis and/or eVTOL, such as airbags and being shockproof, among others [51] are already used in autonomous land vehicles. However, more preventions are still needed, be they for [35,44] collisions or equipment or system failures. There is also an extensive research effort on vehicle separation, communication, and airspace integration. However, these are not mentioned in detail as they are outside the scope of the main focus of this research.

One of the most-cited characteristics in the literature is resilience [40,48,52,58]. Resilience in engineering derives from the resilience of material science and is characterized by the material’s ability to return to its original state after deformation. In the figurative sense, it means the “ability to adapt or recover” [61]. A resilient security architecture is presented in [40], whose proposal is to identify GPS spoofing attacks after being subjected to a failure and manage, through historical data techniques, to return to the current state of the network. However, this work focuses on possible GPS spoofing, not dealing with the case of an advanced GPS attack in which the vehicle has already been subjected to an attack that has obtained partial or total access to the vehicle’s network.

Regarding the problems of location information privacy attacks, such as the GPS in the attacks mentioned above, a mechanism is proposed in [49]. This study presents an obfuscation mechanism for concealing accurate information by transmitting modified location parameters, either after diluting their accuracy or fabricating misleading trajectories for a known eavesdropper. These two broad categories are defined as obfuscation techniques based on attenuation and deception, respectively. The authors also present a taxonomy of 3D obfuscation mechanisms supported by formal descriptions of underlying operators. The mechanisms proposed by the authors were evaluated in theory and practice through a customizable obfuscator implanted in an unmanned aerial vehicle with a GPS. Despite being tested in a UAV, this mechanism can be studied to be reproduced in eVTOLs and replicated in air taxis.

To allow air taxis to perform free-flight operations autonomously and safely, one study [53] presents a computational guidance algorithm for collision avoidance. The approach uses the Markov decision process to solve the Monte Carlo tree search algorithm. To visualize the results, the authors used a high-density free-flight airspace simulator created precisely to test the performance of the proposed algorithm. According to [53], the numerical results show that the proposed algorithm has fewer conflicts and collisions in the air than the optimal reciprocal collision avoidance (ORCA) [62], a state-of-the-art collision-avoidance strategy.

Autonomous air taxis and normal architecture ones will work in a traditional and open environment, with random possibilities of events. Therefore, these architectures have degrees of freedom. Different studies found in the literature agree that these vehicles will experience different eventualities during their use. As a result, these vehicles will be exposed to different security threats. In this sense, vehicles cannot be subject to human control, such as the crew or the airport [43], as safety problems in this sense are already known. According to [54], at the system level, secure computing and communication are required for critical security functions such as flight perception and navigation. The former is usually accomplished by implementing state-of-the-art machine learning techniques [9,12,34,54,55,60]. An example is presented in [12], where the authors present a cloud-based geospatial intelligence system that intermediates information between a set of intelligent agents through a reinforcement learning approach to determine optimal flight policies through these airways. These policies can take into account a variety of factors (wind, precipitation, communication, etc.) that affect the UAV’s path-following capabilities. However, although these techniques are sophisticated, they can be tampered with during operation, for example, by unauthorized substitution or intentionally triggering misclassifications.

Although the terms prevention and prediction are related, in this work, prevention was treated in the previous paragraphs as the act of preventing accidents from happening. In the next section, techniques for forecasting are discussed, i.e., forecasting accidents, failures, etc., that may or may not occur.

### 3.3. Forecasting Accidents

There are techniques for predicting accidents involving fatalities [34] and for hazard recognition and risk assessment in open and non-predictive environments to support decision making and action selection for UAS [63]. Machine learning is the technique that has received the most attention from researchers to predict accidents in air taxis and eVTOLS, as can be seen in Table 4. According to [14], machine learning and artificial intelligence provide more-accurate self-diagnostics to help anticipate problems and report them before an error is noticed, as well as to provide recommendations for repairs and to reduce costs. Further, according to [14], data analysis and predictive maintenance allow failure detection long before it occurs and provide recommendations to ensure repairs are carried out when needed, resulting in less downtime and reduced operating costs.

A technique for deriving classification models to predict conditions that increase the probability of air crashes involving fatalities and serious injuries is presented in [34]. The study comprises machine learning classification techniques, including decision trees, K-nearest neighbors, support vector machines (SVMs), and artificial neural networks (ANNs) that are applied to datasets derived from original data obtained from Federal Aviation Administration (FAA) aviation accident and incident records from 1975–2002. Accident data are filtered to focus on FAA Part 91 (General Aviation) accidents involving manufactured, fixed-wing, and powered aircraft. According to the authors, ANNs produce the most accurate prediction levels for fatal accidents and accidents involving severe injuries.

In addition to machine learning techniques, Bayesian belief networks and object-oriented BNNs also appear in the literature. In [35], the collision probability in the air was calculated using a Bayesian belief network (BBN). BBN was used to model system security of alternative geofencing configurations for small vehicles. UASs deployed in precision agriculture were used as a case study to assess the probability of a mid-air collision in uncontrolled airspace.

Only one study [54] mentions the encryption of these types of vehicles. According to the authors, the essential class of hardware security services includes encryption accelerators and mechanisms to protect data, keys, and identities to provide efficient and secure certificate-based authentication and attestation of networked devices. This and other works in the literature do not propose encryption techniques, despite the importance of encryption to avoid different types of attacks, such as GPS spoofing.

### 3.4. Tests

The tests for air taxis are varied, and some authors choose to use real accident data [34], while others use simulation software [9,35]. Table 5 presents some of the types of tests found in the literature.

In [34], we used a dataset of FAA aviation accident and incident records from 1975–2016 to test the application of different machine learning techniques to derive classification models to predict conditions that increase the probability of aircraft accidents involving fatalities and serious injuries. According to the authors, FAA data are divided into accident and incident records and cover FAR Parts 91 (GA), 121 (commercial aviation), 135 (air taxis), ultralights (103), parachute operations (105), helicopters (133), agricultural operations (137), and flight schools (141).

In [16], a multi-agent system was used, which, according to the authors, has the advantage of the possibility of creating and simulating multiple vehicles (or agents) that act independently, as would happen in real situations. This tool uses the Netlogo language and allows the modeling of multiple UAVs with stochastic trajectories created in a decentralized way, verifying how the model deals with conflicts and obstacles [16]. Safety-driven behavior management, on the other hand, has as its main focus the modeling of situations and problems of knowledge representation in the context of situational risks, and can be customized depending on the research objective [63]. Finally, the systolic FLS architecture proposed in [13] was designed and tested using MATLAB and VHDL to interface with five Lidar sensors and five ultrasonic sensors using an Intel Altera OpenVINO FPGA board.

Although there are different types of [33,34] software and databases available in the literature that can be used for simulation, the literature still lacks more complex tests. The literature will still have to advance some steps to find tests with real vehicles, using architectures and techniques in practice. Only by taking this big step forward can we consider the popularization of air taxis in all parts of the world.

### 3.5. Security Systems and Architectures

Different systems and architectures with different goals are found in the literature. Not all of them are specific to air taxis, such as the Drone Net [48]. However, they can be used as study architectures to replicate security techniques in air taxi architectures, given the high ratio of UAVs to eVTOLS. Table 6 presents some of the architectures and techniques found through this literature review.

Security architectures aimed at UAVs, such as Drone Net [48], STUART [40], and HAMSTER [41], use different types of approaches. HeAlthy, Mobility, and Security-based data communication archiTEctuRe (HAMSTER) is a data communication architecture designed for improving mobility, security, and safety of the overall system [41]. HAMSTER is divided into three categories: (1) flying HAMSTER for air vehicles, (2) swimming HAMSTER for water vehicles, and (3) running HAMSTER for land vehicles. HAMSTER integrates four platforms: (1) Nimble, (2) NCI, (3) NP, and (4) SPHERE, where the SPHERE platform is focused on architectural security and safety issues. Security and safety Platform for HEteRogeneous systEms (SPHERE) is HAMSTER’s safety and security platform. Unmanned vehicles have different peripherals and modules, so they require different levels of safety, thus leading to the need to classify modules according to their importance and criticality. According to [41], modules are categorized as primary or secondary. Primary modules have the crucial components for the aircraft to fly, be aware of its location, and be able to make an emergency landing safely. These modules are the GPS receiver, autopilot, flight plan, radar, etc. The authors carry out architecture validations through concrete implementations of HAMSTER modules.

Despite significant efforts with HAMSTER, the authors do not consider aspects of resilience for autonomous vehicles, unlike STUART [40], which is a resilient architecture to dynamically manage unmanned aerial vehicle networks under attack. This architecture is one of the only ones that works with the concepts of safety and security together, using decision techniques through a state machine and resilience through restoration techniques through historical data saved in a reference base. It is worth mentioning that one of the concerns in the design and development of UAV systems has been to ensure safety requirements. However, since these vehicles communicate with external entities, some architectures designed to provide safety requirements can present security flaws, and security requirements can have safety flaws. Therefore, it is necessary to investigate and treat safety and security together in a UAV architecture. In this sense, STUART is the only architecture concerned with both insecurities. However, the architecture is still in the simulation testing phase and has not been tested on a real avionics architecture.

Likewise, [68] presents a container framework that proposes resilient control against DoS attacks for real-time UAV systems using containers. Container technology is open source and offers software isolation [69,70] and abstraction of many features in the Linux kernel. System isolation using containers leads to less execution overhead, less memory usage, and a smaller footprint. Compared to virtual machines (VMs), containers have advantages in lightness [68] because they share the host kernel, while a virtual machine needs a hypervisor to communicate with the kernel. In terms of efficiency, a container is generally tens of MBs, while a VM easily exceeds a GB. ContainerDrone provides support mechanisms for three system resources, namely the central processing unit (CPU), communication channel, and memory, and also offers resilient control against DoS attacks for real-time UAVs. The ContainerDrone framework has proven to be reliable in protecting against DoS attacks launched within the container by limiting attacker access to three critical system resources: the CPU, memory, and communication channel. Experiments have shown that the proposed structure can be effective against various types of container control environment (CCE) DoS attacks. In this study, the authors did not consider physical component failures, software failures caused by bugs and/or logic failures, or any attack other than DoS.

In one study [12], a cloud-based geospatial intelligence system called BRECCIA is presented. According to the authors of [12], this intelligent system intermediates information (such as roads, topography, temperature, wind, precipitation, GPS, V2X communication, buildings, towers, etc.) between a set of intelligent agents. BRECCIA has a reinforcement learning approach to determining optimal flight policies through these airways, with these policies taking into account a variety of factors (wind, precipitation, communication, etc.) tracking the path of the UAV. Unlike STUART, which is focused on safety and security issues, the BRECCIA system focuses on safety challenges.

The authors of [7] present a scarecrow architecture to be used as a basis for an airspace system in which autonomous aircraft will be controlled by autonomous controllers. A scarecrow architecture utilizes geo-enclosed areas with manning restrictions as well as aircraft volume restrictions from which these geo-enclosed areas are dynamically computed. According to the authors in [7], the safety margin of the dynamic structure is continuously calculated to provide guidance to the control algorithms that regulate airspace loading. Scarecrow Architecture, BRECCIA, HAMSTER, and STUART are four examples of concise, drill-down architectures for air taxis, each with its own goals and objectives as described.

The most widespread architecture in the literature, not only in terms of references but also in terms of technical software and hardware evolutions, is NASA’s vehicle [11,32,54,66,67]. NASA has contributed efforts in the development of air taxi aircraft designs. These efforts are a contribution for other researchers to reuse definitions and technologies as a support for the development of their research [32]. NASA’s vehicle encompasses different technologies such as propulsion architectures, highly efficient and quiet rotors, and aircraft aerodynamic performance and interactions. According to [32], the configurations adopted are generic and are intentionally different in appearance and design details from prominent industry arrangements. These UAM concept aircraft have already been used in numerous engineering investigations. In this regard, the authors of [66] present a research effort for the NASA Ames Research Center. The authors present a software prototype, called NAS Integrated Collaborative Planning System (NICoPS), to support all NAS users in planning their operations in collaboration with traffic managers and licensing personnel. NICoPS supports a variety of airspace users, including emerging markets with different types of procedures and vehicles. In [11], SWIM is presented, a platform that implements a publish/subscribe model that allows many other users to subscribe to data at once. The SWIM program aims to support information-sharing among NAS stakeholders by providing a communication infrastructure and architectural solutions to identify, develop, provision, and operate reusable, shareable services. According to [66], the future NAS must be shared equitably between all vehicles used for various missions that present different security and planning challenges.

## 4. Conclusions

This section discusses current approaches in progress in the literature and open gaps in general. The first point to be discussed is about recent accidents, because, as far as we know, there are no studies that use up-to-date databases considering the evolution of aviation in recent years. Aviation has progressed, and with that, the causes of accidents have also changed, so using recent databases will bring an accurate view with respect to air taxis through new sets and prediction models.

As discussed throughout the results, one of the main concerns of current pilots is detailed information about the weather during flights. Examples of this information are wind gusts, thunderstorms, unexpected air events, bad weather alerts, etc. Although systems with this information already exist, state-of-the-art systems with high accuracy of real-time information are still lacking. When the sky is clear, this gap involves challenges in deploying observation sensors and combining these sensors with meteorological models so that pilots can observe this information accurately and in real-time.

Once air taxis are established in the real world, they will share airspace with other vehicles, including drones, commercial planes, and other air taxis. As far as we know, there is still no in-depth research focusing on the safe planning of this shared space between air taxis. Although not the main focus of this work, the gap in regulatory and safety efforts for these vehicles is worth mentioning. The lack of patronization is not only in regulatory laws but also in architecture development. Each company or country dictates its own rules, which can generate future problems between different regulations. There are already studies focused on solving these problems, but no worldwide organization is working towards this standardization.

The sophistication of these vehicles will increasingly demand improvement to internal decision-making capabilities, as air taxis will have to continually adapt to missions and detect unexpected internal problems or external hazards. These decisions must consider the mission objective and adjust the choice of actions to achieve it according to random events related to the mission context, the health of the entire system, the handling of risk, and mission security. However, several risks are involved in autonomous avionics systems, such as cybersecurity issues, safety, airworthiness, etc. Ongoing efforts focus on developing mathematical and predictive models using algorithms and different techniques to support this transition of vehicles until we reach full autonomy. However, it is impossible to reach a level of autonomy and absolute security where risks do not exist. Flaws and errors will always exist, and therefore, studies to update the community, whether public or research, and practical and theoretical studies will always be valid. Air taxi safety and security issues will always go hand-in-hand to provide the highest possible level of safety assurance for passengers, the environment, and society. Finally, one of the existing gaps that is not addressed in the literature is energy increase after the implementation of these vehicles. If this means of transport is intended for a very small group of users, air taxis will occupy a similar place as the private helicopters that have been used for years. Therefore, it is necessary to analyze the increase in energy that will occur with the use of these vehicles.

The main future work of this study will be the ideation and implementation of an avionics architecture for air taxis. STRAUSS is a new resilient, robust, and fault-tolerant architecture for eVTOLs that operate in adverse, intentional, or unintentional conditions. This architecture will increase cooperation between air vehicles due to a reliable architecture based on blockchain. In addition, it will be composed of an intelligent and innovative system to help aircraft detect real-time weather characteristics (including the presence of wind gusts), even in difficult situations where there are no clouds. All proposed algorithms will be implemented on specific hardware (real drone, 3DR solo, etc.) and intelligently share information between air taxis to minimize the energy consumption induced by STRAUSS. Future work includes implementing and comparing different techniques for each of the parts that will make up STRAUSS. 

## Figures and Tables

**Figure 1 sensors-22-06875-f001:**
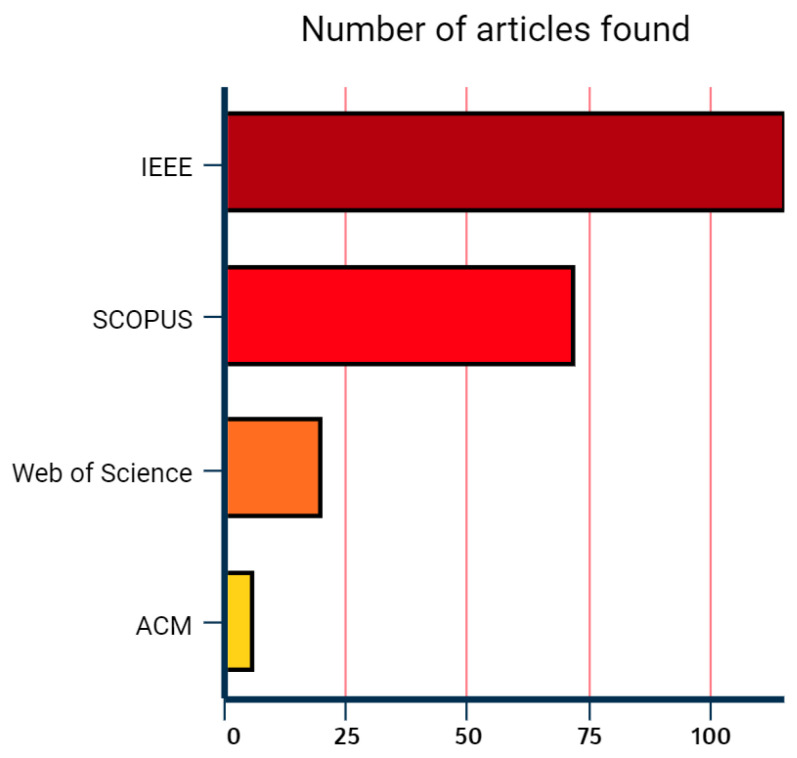
Number of articles found in databases during the first selection.

**Figure 2 sensors-22-06875-f002:**
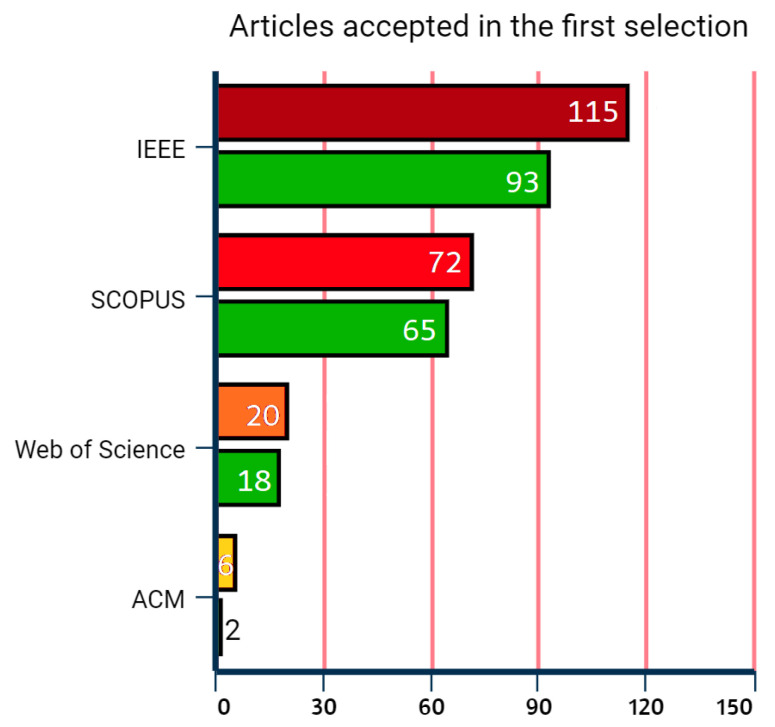
Number of articles accepted during the first selection.

**Figure 3 sensors-22-06875-f003:**
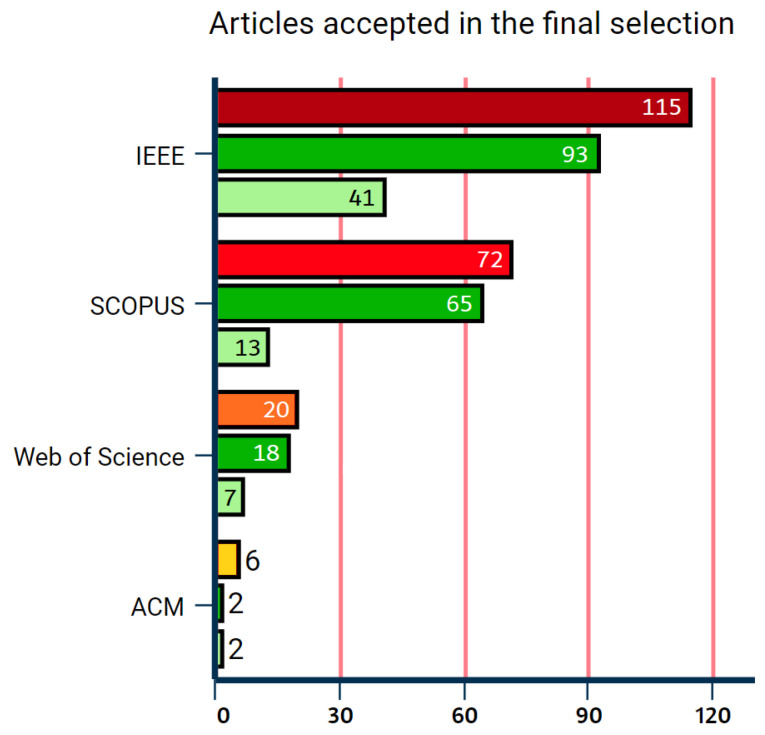
Number of articles accepted in the final selection.

**Table 1 sensors-22-06875-t001:** Literature review studies of air taxis.

Objective of the Systematic Review for Air Taxis	Study
Urban air mobility	[18,21]
Demand for services	[18,19,20]
Safety and security aspects.	This study

**Table 2 sensors-22-06875-t002:** Main reasons for air taxi accidents.

Cause of Accident	Study
Human error	[33,34,35,36,37,38,39]
Equipment failure	[33,35,36,40,41,42]
Crew and passengers	[33,43]
Airport	[33]
Lack of data	[30,37]
Weather	[14,28,30,33,35,37,44,45]
Collision	[35,44]
System attacks	[35,40,42,46,47]

**Table 3 sensors-22-06875-t003:** Accident prevention techniques.

Technique	Study
Landing gear, subfloor, shockproof, airbags and seat	[51]
Active crash protection system	[51]
Deployable energy absorber	[51]
Resilience	[40,46,48,52,58]
Markov decision process	[53]
Monte Carlo tree search	[53]
3D obfuscation	[49]
Optimization	[54,55,56,57]
Multi-agent system	[16]
Machine learning	[9,12,34,54,55,59,60]

**Table 4 sensors-22-06875-t004:** Main techniques to forecast accidents.

Technique	Study
Machine learning	[14,34,36,54,63,64]
Bayesian belief network	[35]
Object-oriented BBN	[35]
Gas models	[35]
Reliability diagram	[36]
Artificial intelligence	[54]
Encryption	[54]
Trajectory-based operations overview	[65]

**Table 5 sensors-22-06875-t005:** Tests for air taxis.

Type of Test	Study
Accident data	[34]
GPS fault emulation	[35]
Multi-agent system	[16]
Airborg SITL simulation	[9]
Non-specific simulation	[40,41,46]
Safety-driven behavior management	[63]
Systolic FLS architecture	[13]

**Table 6 sensors-22-06875-t006:** Security systems and architectures.

Architectures	Study
Drone Net	[48]
NASA vehicle	[11,32,54,66,67]
Volocopter eVTOL	[17]
STUART	[40,46]
HAMSTER	[40,41]
Container framework	[68]
Modular data modeling	[45]
Based airspace management system	[8]
Straw-man architecture	[7]
TOL modes of the main modern flying cars	[10]
SWIM	[11]
BRECCIA	[12]
Systolic FLS architecture	[13]
Framework for level of autonomy	[15]

## Data Availability

Not applicable.

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
