# Peer review of "Security and Safety Concerns in Air Taxis: A Systematic Literature Review"

_sensors, 2022, doi:10.3390/s22186875_

Round 1

Reviewer 1 Report

I would suggest to analyse what passenger flow could be served by Air Taxis in reality. In very big aglomerations (e.g. Sao Paulo) the majority of people use public overland or undergroud transport modes. There are missing scenarios to create a new mobility system including Air Taxis. If this mean of transport is dedicated for a very low group of users the Air Taxis will take a place similar to private helicopters which have been used for years. If Air Taxis becomes popular one have to analyse what increase of energy will occure. This is a question of security of the new solution and this topic is not concidered in the article

Author Response

Carrying out a conceptual study for the applicability and effectiveness of using air taxis on demand is a computational challenge, mainly due to the conflict between modeling a large urban area and the desire to have a high-resolution model that can be used for an accurate prediction of transport mode usage by introducing on-demand air taxi service. We did not include this type of analysis mainly because this subject was not addressed in the analyzed articles. However, we inserted it as a gap in the conclusions to guide future work. As our work is a systematic review and the topic was not discussed in the reviewed articles, we had not previously mentioned it despite being very important and relevant. Thanks for the suggestion!

Reviewer 2 Report

The study aims  to carry out an extensive Systematic Literature Review seeking to present the main modern advances, such as the techniques, architectures, and research carried out around the world focused on these types of vehicles.

there are major revision 

1-In abstract, More than 210 articles were individually reviewed between 12 2015 and January 2022. So why referencs only are 71, justify the rest of papers 

2-it is great if authors make a table in the introduction section to  summarise the existing  literature reviews 

3-some realted work are missing such as Zero-padding and spatial augmentation-based gas sensor node optimization approach in resource-constrained 6G-IoT paradigm

4-tables need more explanation and details for each study such as highlighted , focus, .....

5-conclusion should be one paragraph summaries your findings 

6- in 3. Results and discussion than 4. Synthesis of results while section 3 is empty. why justification or something is missing 

7-What is the full form of UAV and etc. 

Author Response

1-In abstract, More than 210 articles were individually reviewed between 12 2015 and January 2022. So why references only are 71, justify the rest of papers

Dear reviewer, 210 articles were individually reviewed, however, only 71 were cited because the others did not pass all the acceptance criteria or fit one of the rejection criteria, performed during the systematic review. The acceptance criteria used were: Studies that address aspects of safety and/or security in air taxis; that use techniques and/or algorithms of safety or security in flying cars; Studies that do not address safety or security address innovations for flying vehicles. The defined exclusion's criteria were: Studies that do not have a clear structure of objectives and results; Studies that have no relevance to the main theme; Studies that address flying vehicles but do not address safety or security aspects; Incomplete studies; Studies that are not in English language; Studies that are not available for download. These criteria can be found in the chapter Systematic Literature Review Method - planning.

2-it is great if authors make a table in the introduction section to summarize the existing literature reviews

Suggestion accepted

3-some realted work are missing such as Zero-padding and spatial augmentation-based gas sensor node optimization approach in resource-constrained 6G-IoT paradigm

Suggestion accepted

4-tables need more explanation and details for each study such as highlighted , focus, .....

Suggestion accepted

5-conclusion should be one paragraph summaries your findings 

Dear reviewer, in the conclusion we insert a summary of our findings that are gaps that may guide future works, we discuss current approaches in progress in the literature and open gaps in general. An example of a finding was about recent accidents, as far as we know, there are no studies that use up-to-date databases because considering the evolution of aviation in recent years. Aviation has progressed, and with that, the causes of accidents have also changed, so using recent databases will bring air taxis an accurate view through new sets and prediction models. In addition, in the conclusion we also used the space to discuss other findings on each topic addressed in the results.

6- in 3. Results and discussion than 4. Synthesis of results while section 3 is empty. why justification or something is missing 

Suggestion accepted, we actually forgot this subsection in the text. Thank you so much!

7-What is the full form of UAV and etc. 

In this study we analyze works considering electric vertical take-off and landing aircraft.

Round 2

Reviewer 2 Report

Authors addressed my comments very well